

# The behavior and welfare of neglected species: some examples from fish and mammals

Syed S. U. H. Bukhari[1,2], Rebecca S. V. Parkes[1,2,3], Lynne U. Sneddon[4] and Alan G. McElligott[2,5]

[1] Department of Veterinary Clinical Sciences, Jockey Club College of Veterinary Medicine and Life Sciences, City University of Hong Kong, Kowloon, Hong Kong, China
[2] Centre for Animal Health and Welfare, Jockey Club College of Veterinary Medicine and Life Sciences, City University of Hong Kong, Kowloon, Hong Kong, China
[3] Large Animal Medicine and Surgery Department, School of Veterinary Medicine, St. George's University, True Blue, St. George's, Grenada, West Indies
[4] Department of Biological & Environmental Sciences, University of Gothenburg, Gothenburg, Sweden
[5] Department of Infectious Diseases and Public Health, Jockey Club College of Veterinary Medicine and Life Sciences, City University of Hong Kong, Kowloon, Hong Kong, China

Corresponding authors
Syed S. U. H. Bukhari,
habukhari2-c@my.cityu.edu.hk
Alan G. McElligott,
alan.mcelligott@cityu.edu.hk

## ABSTRACT

Animal welfare is the state of an animal's body and mind and the level to which its requirements are satisfied. Animal welfare is affected by human decisions and actions. Numerous decisions concerning animals are driven by human desires to enhance their own lives, and some of these decisions may be influenced by self-interest or a strong emphasis on economic factors. How to assess the welfare state of animals is a central issue in animal welfare science. Two critical questions can be used to address animal welfare: first, is the animal healthy, and second, does the animal have what it needs? Both of these questions can potentially be answered using the study of animal behavior. The development of behavioral methodologies is crucial for evaluating welfare in contexts where concern for animal welfare is often highest, such as on intensive modern farms and sites where working animals are used. Herein, we discuss animal welfare by focusing on some of its major concepts and explanations. Later, to illustrate key aspects of animal welfare, we chose to examine the information that is available for some 'neglected' livestock species, which are commercially important on a global basis and found in large numbers: buffaloes (*Bubalus bubalis*), camels (*Camelus dromedarius*), donkeys (*Equus asinus*), mules (*Equus asinus* × *Equus caballus*), and lumpfish (*Cyclopterus lumpus*). We chose these species because there are major ongoing concerns about their welfare, and more research is required to help solve the various problems. Overall, there are strong imbalances in terms of the species that are usually studied in terms of animal welfare research, and we call for greater attention to those that have traditionally been neglected.

## INTRODUCTION

Animal welfare refers to the physical and mental state of an animal and the degree to which its needs are met (*Dawkins, 2021*). For example, for housing, a black color water tank may improve lumpfish (*Cyclopterus lumpus*) welfare as studies have suggested that lumpfish prefer a black background rather than the typical blue or light grey aquaculture tank color (*Garcia de Leaniz et al., 2015*). Animal welfare thus encompasses the mental and physical wellbeing of animals, as well as their physiology, functions, and adjustments to diverse environments and various challenges (*Mellor & Beausoleil, 2015*). Welfare can be inferred from many factors including the relationships between animals, their surrounding environment, quality of life, and adaptability (*Dawkins, 2021*).

Animal welfare science is concerned with assessing how an animal is dealing with its surrounding environment (*Baciadonna & McElligott, 2015*; *Dawkins, 2021*). Animal ethics, as opposed to animal welfare, concerns human-animal relations, how to apply animal welfare science, and how people should treat animals (*Beauchamp & Frey, 2011*), as human actions have a direct impact on animal welfare. For example, stockpersons who hit, slap, or yell can cause milk letdown problems, higher heart rates, blood cortisol, and retained milk in buffalo and cattle (*Rushen, De Passille & Munksgaard, 1999*; *Napolitano et al., 2013*). Ethical considerations are important for legislation and research policies. Animal ethics issues are often contentious because there is no global consensus on how we should treat animals (*Beauchamp & Frey, 2011*; *Hvitved, 2019*). Thus it follows that researchers should take care of the welfare of animals, irrespective of their utility or function, as well as their rights as living and sentient organisms (*Beauchamp & Frey, 2011*; *Hvitved, 2019*).

The Five Freedoms was the first widely recognised evidence-based framework to encompass key aspects of animal welfare (needed to minimize welfare risk) in a single model (*Arndt, Goerlich & van der Staay, 2022*; *Miller & Chinnadurai, 2023*). It has been used as a framework for addressing animal welfare, especially in the context of farm animals (*Arndt, Goerlich & van der Staay, 2022*; *Miller & Chinnadurai, 2023*). This framework includes freedom from hunger (and malnourishment) and thirst by ready access to a nutritious diet and fresh water; and freedom from discomfort (lack of physical as well as thermal distress) by good shelter and resting places (*Mellor, 2016*). The Five Freedoms also include freedom from pain, injury, and disease by preventative measures or quick testing and treatment; freedom to express normal behavior by providing appropriate space; and freedom from fear and stress by ensuring living conditions, diagnosis and cure which nullify mental distress. Since their development, the Five Freedoms have contributed greatly to improving animal welfare standards (*Mellor, 2016*). Further, the Five Freedom model has recently expanded to include human-animal interactions (*Mellor et al., 2020*).

In a similar manner to the Five Freedoms, the Five Domains of animal welfare (Table 1) relate to an animal's overall welfare state as understood in terms of its affective experiences (*Dawkins, 2021*). These domains similarly include nutrition, environment, health, behavior, and mental state. The Five Domains model, allows for the consideration of positive experiences (as well as the absence of negative ones), which may improve welfare

**Table 1  Five domains model of animal welfare (*Mellor & Beausoleil, 2015*).**

| Nutrition | | Environment | | Health | | Behavior | | Mental states | |
|---|---|---|---|---|---|---|---|---|---|
| Positive | Negative | Positive | Negative | Positive | Negative | Positive | Negative | Positive | Negative |
| Availability and opportunity to drink or live in clean water and eat an appropriate diet | Restriction on water and food intake. Unavailability of quality food and water. | Availability of species-specific environmental conditions (space, air, odour, light, noise, temperature *etc.*). | Unsuitable environments (thermal extremes, close confinements, pollution, inappropriate light, odor, noise *etc.*). | Absence of diseases, injuries and functional impairments. Appropriate body condition and proper fitness. | Presence of acute and chronic diseases, poor body condition, lack of medical facilities *etc.* | Ability to express species-specific behavior, availability of enriched and engaging choices, allowed for sufficient rest and sleep. | Restricted choices, limitations on sleep and rest, restrictions on threat avoidance, escape or defensive activity. | Pleasure to drink and eat. Thermal, physical, respiratory, olfactory, auditory and visual comfort. Positive mental states. | Thirst and hunger. Thermal (chilling, overeating), physical (joint pain, skin irritation, muscle tension and stiffness), respiratory (breathlessness), olfactory, auditory (impairment and pain), and visual discomfort. Negative mental states such as pain, fear or stress. |

(*Mellor & Beausoleil, 2015*). The model provides a more comprehensive method for considering welfare (*Mellor & Beausoleil, 2015*). In the context of poor welfare, the first four domains focus on internal physiological and pathophysiological disturbances due to nutritional, environmental and health-related problems (domains 1–3), and on external physical, biotic and social conditions in the animal's environment that may limit its capacity to express various behaviors or may otherwise pose significant challenges (Domain 4) (*Mellor & Beausoleil, 2015*). Once such internal and external factors are assessed, their anticipated affective consequences are assigned to the fifth 'mental' domain, and it is these experiences that determine the animal's welfare state, Table 1 (*Mellor & Beausoleil, 2015*).

Most animal welfare investigations have been concerned with the alleviation of pain, stress, and suffering (*Briefer, Tettamanti & McElligott, 2015*; *Spence, Osman & McElligott, 2017*; *Dawkins, 2021*). However, promotion of the positive animal welfare states has become increasingly important in animal welfare over the past 20 years (*Briefer, Tettamanti & McElligott, 2015*; *Laurijs et al., 2021*; *Dioli, 2022*; *Bukhari et al., 2023*). Positive animal welfare and related concepts, such as "life worth living," happiness, contentment, and good welfare, are becoming increasingly prevalent in the field of animal sciences (*Mellor & Beausoleil, 2015*). Examples of practices that can enhance positive animal welfare include providing buffalo with access to water pools and mud for wallowing (*Napolitano et al., 2013*), or giving camels sandy floor bedding (*Ahmad et al., 2010*), in modern farming systems. Positive animal welfare highlights the benefits of providing animals with positive experiences, in addition to reducing their suffering. These ideas are closely linked to the Five Domains model (*Mellor & Beausoleil, 2015*), which emphasizes the importance of considering an animal's physical, emotional, and mental states when assessing their welfare.

To illustrate key aspects of animal welfare, here we focus on four 'neglected' livestock species, which are very important commercially and found in large numbers across the

world: terrestrial livestock such as buffaloes (*Bubalus bubalis*), camels (*Camelus dromedarius*), donkeys (*Equus asinus*) and mules (*Equus asinus* × *Equus caballus*), and one aquatic species, the lumpfish (*Cyclopterus lumpus*). There are many concerns about their welfare, yet the literature is very limited. The mammal species included are important because they typically inhabit harsh or hot environments (*Bukhari et al., 2022*; *Chikkagoudara et al., 2022*; *Dioli, 2022*), and have enormous potential to assist humanity in dealing with the challenges of climate change, as temperature of our planet is rising (*NASA USA, 2023*). Buffaloes, camels, and donkeys can thrive in hotter climates (*Bukhari et al., 2022*; *Chikkagoudara et al., 2022*; *Dioli, 2022*). Buffaloes and camels thrive in hot climates and can reproduce, produce milk, and provide meat (*Marai & Haeeb, 2010*; *Dioli, 2020*, *2022*). Donkeys, on the other hand, are utilized for work in hotter regions and to assist humans with transportation (*Bukhari & Parkes, 2023*; *Bukhari et al., 2023*). Furthermore, donkeys also serve as production animals for high quality meat, nutritious milk, and their products are used in cosmetics (*McLean & Navas Gonzalez, 2018*; *The Donkey Sanctuary, 2023*). Our objective was to highlight the neglect of these livestock species (buffalo, camel, donkey, mule, and lumpfish) in terms of the number of welfare publications available. There are numerous livestock species that have not received adequate attention to date, such as ducks (*Anas platyrhynchos*) and turkeys (*Meleagris gallopavo*), (*Eurogroup for Animals, 2023*), and they also deserve renewed attention. We hope that this review will serve as a catalyst for future research on the welfare of these and other species, which is greatly needed.

## METHODS

Based on our extensive knowledge of the literature, we selected livestock species that we believed were likely to be relatively neglected in terms of animal welfare research, and then paired them with species (phylogenetical relatives) that tend to feature more commonly in the animal welfare literature. The terms used to conduct the search were: ("Buffalo" "Welfare"), ("Cattle" "Welfare"), ("Camel" "Welfare"), ("Sheep" "Welfare"), ("Donkey" "Welfare"), ("Mule" "Welfare"), ("Horse" "Welfare"), ("Lumpfish" "Welfare"), and ("Salmon" "Welfare"), (Fig. 1). Cattle were chosen as a contrast to buffalo, horses compared to mules/donkeys, sheep compared to camels, and salmon was chosen as a contrast to lumpfish. The following inclusion criteria were used to select the published articles (to show that the selected species in this review are indeed neglected), including Scopus research articles, reviews, case reports, publications ranging from January 2000 to December 2022, and one of the search terms present in title, abstract or keywords of the articles. We restricted our search to scientific literature published between January 2000 and December 2022, to ensure a current and up to date perspective for these topics.

Globally, there are 1.5 billion cattle (*Hegde, 2019*), and 208 million buffalo (*Minervino et al., 2020*), but we found 6,047 articles for cattle and 235 for buffalo. Even though there are seven times more cattle, articles published on buffalo welfare are 26 times less as compared to those on cattle welfare. Similarly, there are 1.2 billion sheep (*Hegde, 2019*) and 33.7 million camel (*Oselu, Ebere & Arimi, 2022*) globally, but there were only 90 articles on camel welfare compared to 2016 for sheep welfare. Even though there are just 3.4 times
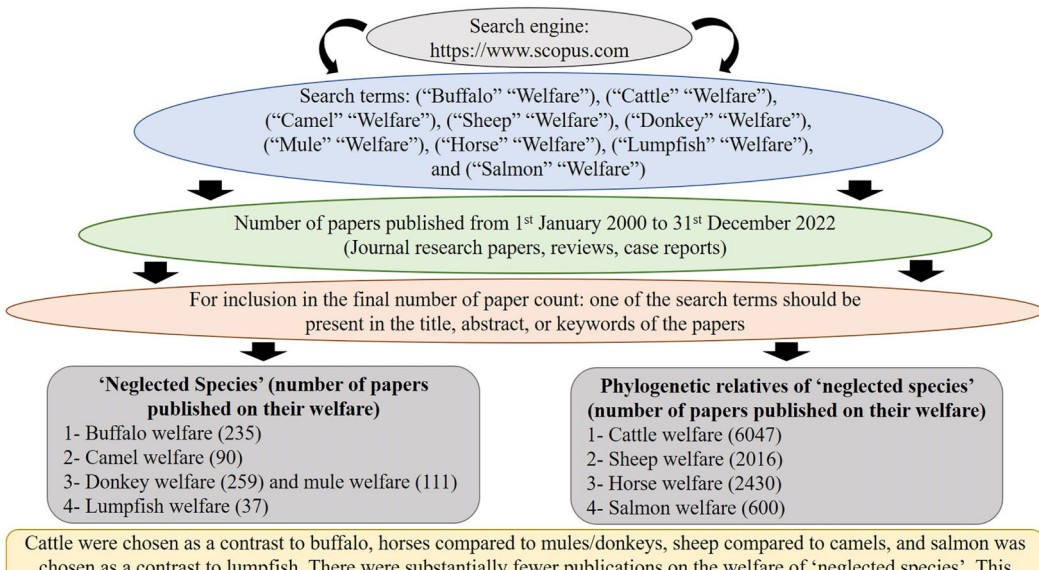

**Figure 1** Overview of the methodology, search engine, search terms, and number of published articles focusing on the welfare of five neglected species and a chosen example of a phylogenetic relative.

more sheep, articles published on camel welfare are 22.5 times less frequent. Moreover, donkeys (global population = 50.5 million) and mules (global population = 8.5 million; 18), (*Norris et al., 2021*), combined have similar populations to those of horses (58.8 million) (*Khadka, 2015*), but the numbers of articles published on donkey and mule welfare were 6.6 times less than those for horse welfare. Therefore, cattle, sheep and horse welfare yielded many more publications compared to buffalo, camel, donkeys, and mules. For the chosen fish species, salmon had 16.2 times the number of publications (600 articles) than just 37 found for lumpfish welfare (Fig. 1). Based on our search results, it is evident that there is a large knowledge gap, which may be due to a Global North bias (at least for the terrestrial species), in a similar manner to what has been found for conservation science (*Watson et al., 2017*; *Nakamura et al., 2023*). Research collaborations are dominated by wealthy nations in the global north, highlighting the need for funders and publishers to reward projects that promote science in the global south (*Nature Editorial, 2023*). Here, we use the phrase "Global North bias" to refer to the tendency for animal welfare studies to primarily focus on animals of interest in the Global North, such as cattle, horses, and sheep. This bias arises from the fact that countries in the Global North typically have greater access to research funding and more established research programs (*Petersen, 2021*). However, we suggest that it is crucial for researchers in these countries to also consider the welfare of other important animals in the Global South, such as buffaloes, camels, donkeys, and mules.

Buffalo and cattle belong to the bovid family and are closely related (*de Rosa et al., 2009b*; *Napolitano et al., 2013*). While cattle are found all over the world, they are mainly concentrated in the Global North (*Hegde, 2019*), where most research is conducted. By

contrast, buffalo have a more widespread distribution in the Global South and thus have not received sufficient research attention (*Minervino et al., 2020*; *Zhang, Colli & Barker, 2020*). Moreover, sheep, although found in both the Global North and the Global South, are more prevalent in the Global North, whereas camels are more common in the Global South (*Dioli, 2020*; *Oselu, Ebere & Arimi, 2022*). However, as both species are ungulates and phylogenetically related (*Gebreyohanes & Assen, 2017*; *Petersen, 2021*), we have chosen to pair them together in this review to highlight the need for greater attention for these neglected species.

Horses and donkeys are both equids. Horses are more commonly found in the Global North (*Khadka, 2015*), often due to their usage in expensive sports, which leads to a greater research focus on horses in those regions (*Bukhari, McElligott & Parkes, 2021*; *Bukhari & Parkes, 2023*). Alternatively, donkeys are frequently used as working animals in the Global South. Despite the differences in where they are most common, both horses and donkeys share similar welfare needs as members of the Equidae family (*Bukhari, McElligott & Parkes, 2021*; *Bukhari & Parkes, 2023*).

Finally, in terms of the species chosen for this review, salmon and lumpfish are fishes that are compared because they are typically farmed together, with lumpfish used to biologically control an external parasite, the salmon louse (*Hvas, Nilsen & Oppedal, 2018*; *Mortensen et al., 2020*). Even though they share the same production environment, welfare studies predominantly focus on salmon, leaving a gap in research on lumpfish welfare (*Hvas, Nilsen & Oppedal, 2018*; *Mortensen et al., 2020*). This rationale explains our focus on these relatively neglected species, which require more future research attention.

## BUFFALO WELFARE

The domestic water buffalo (*Bubalus bubalis*) has a global population of about 208 million (*Minervino et al., 2020*), and their population is rapidly increasing (*Deb et al., 2016*). The number used in production is continuously growing due to increased demand for milk and dairy products such as cheese (*e.g.*, mozzarella in Italy) (Fig. 2) (*de Rosa et al., 2009a*). Asia has the largest population of buffaloes, making up over 97% of the entire world's population (*Deb et al., 2016*). They play a major role in supplying draught power, meat, and milk in 77 countries (*Minervino et al., 2020*; *Zhang, Colli & Barker, 2020*), with the majority used for milk production. There are two subtypes (river and swamp) that originate from the wild water buffalo (*Bubalus arnee*). River buffalo are native to the Indian subcontinent and have spread west as far as Italy, Greece, the Balkans, and Egypt. By contrast swamp buffalo are found throughout south and east Asia, from Bangladesh and India in the west to China's Yangtze River valley in the east (*Zhang, Colli & Barker, 2020*).

Buffaloes have unique morphological characteristics that enhance their ability to thrive in shaded, hot, and humid environments. They have one-eighth the hair density of cattle, allowing heat to dissipate more easily (*Hafez, Badreldin & Shafei, 1955*). Their melanin-stained skin is beneficial for ultraviolet (UV) ray defense (*Hafez, Badreldin & Shafei, 1955*). Swimming and wallowing behaviors are vital throughout the year, but especially during hot seasons to drain away body heat (Fig. 3). The number of sebaceous

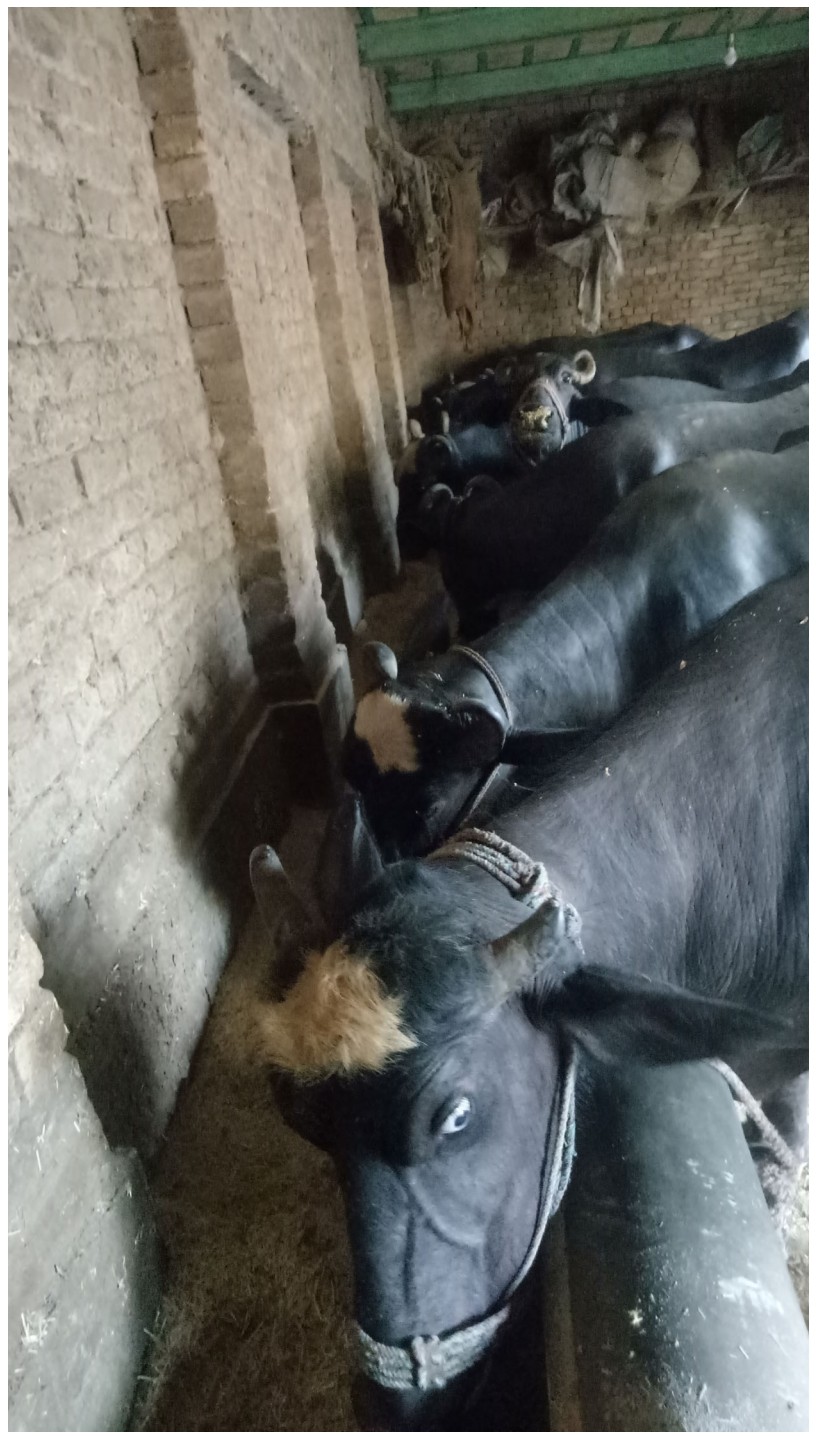

**Figure 2 Buffaloes feeding in a traditional buffalo dairy farm in a rural area of Attock-Pakistan.**
Photo credit: Syed S. U. H. Bukhari.

and sweat glands in buffaloes is less than in cattle, resulting in reduced sweating ability (*Hafez, Badreldin & Shafei, 1955*). Furthermore, buffalo skin is thicker than cattle skin, protecting their body surface from hazardous chemicals as well as mechanical agents,

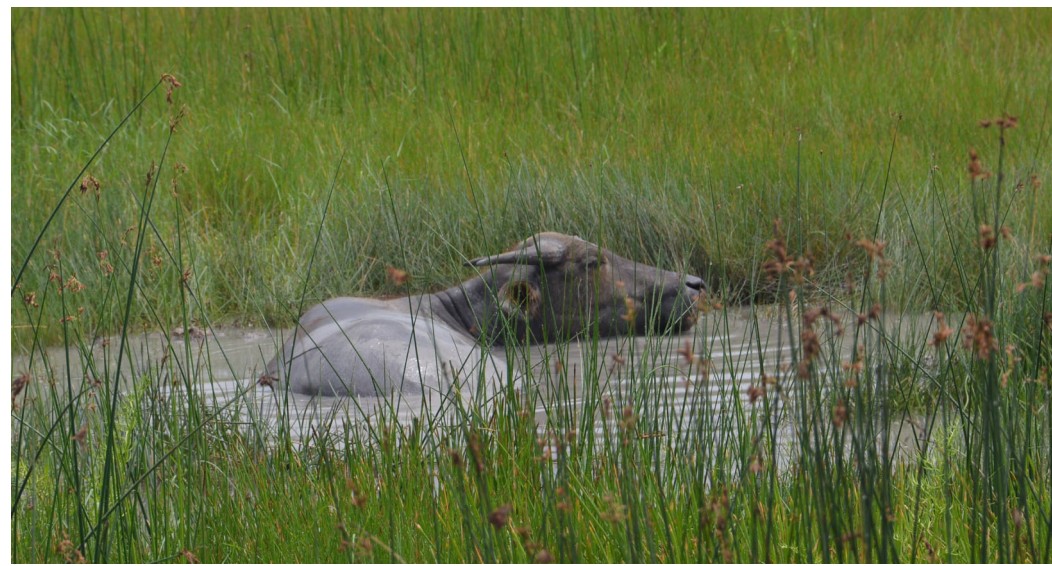

**Figure 3** **Wallowing feral buffalo during summer in Hong Kong SAR, China.** Photo credit: George M. W. Hodgson.                                   

especially when they are in mud and water while wallowing and swimming, respectively (*Hafez, Badreldin & Shafei, 1955*; *de Rosa et al., 2009b*).

Grooming (both social and self-grooming) behavior is critical for body care as well as social structure. Tail swishing is a method for removing flies and irritants from the tail and the back of the body. Body parts within reach are rubbed and licked, whereas inaccessible parts are scratched on readily accessible surfaces or groomed by the other buffaloes after solicitation (*de Rosa et al., 2009b*; *Napolitano et al., 2013*).

## Buffalo welfare issues and limitations for production

Before being domesticated, buffaloes were able to roam freely in the wild (*Deb et al., 2016*; *Zhang, Colli & Barker, 2020*), and used to spend most of their time resting, feeding, and ruminating. Bathing and wallowing are two main buffalo behaviors used to regulate body temperature and protect against ectoparasites in extensive farming (*Mora-Medina et al., 2018*). Buffalo welfare is generally expected to be better in extensive farming because the animals are free to express natural behaviors *e.g.*, wallowing (Fig. 3) and browsing (Figs. 4A, 4B), (*Mora-Medina et al., 2018*). Being able to engage in natural behaviors allows animals to fulfill their own needs. A wide range of natural behaviors are linked with positive emotions, and their performance directly enhances animal welfare (*Špinka, 2006*; *Browning, 2020*). Furthermore, engaging in natural behavior has long-term benefits for animals, including enhanced ability to deal with social and physical problems (*Špinka, 2006*; *Browning, 2020*). However, conventional farming methods are being replaced by more intensive systems designed for dairy cattle, without access to water or mud for wallowing (*Napolitano et al., 2013*).

Intensive farming has exposed buffaloes to changing environments that introduce psychological and physical stressors previously unknown to them, such as confinement,

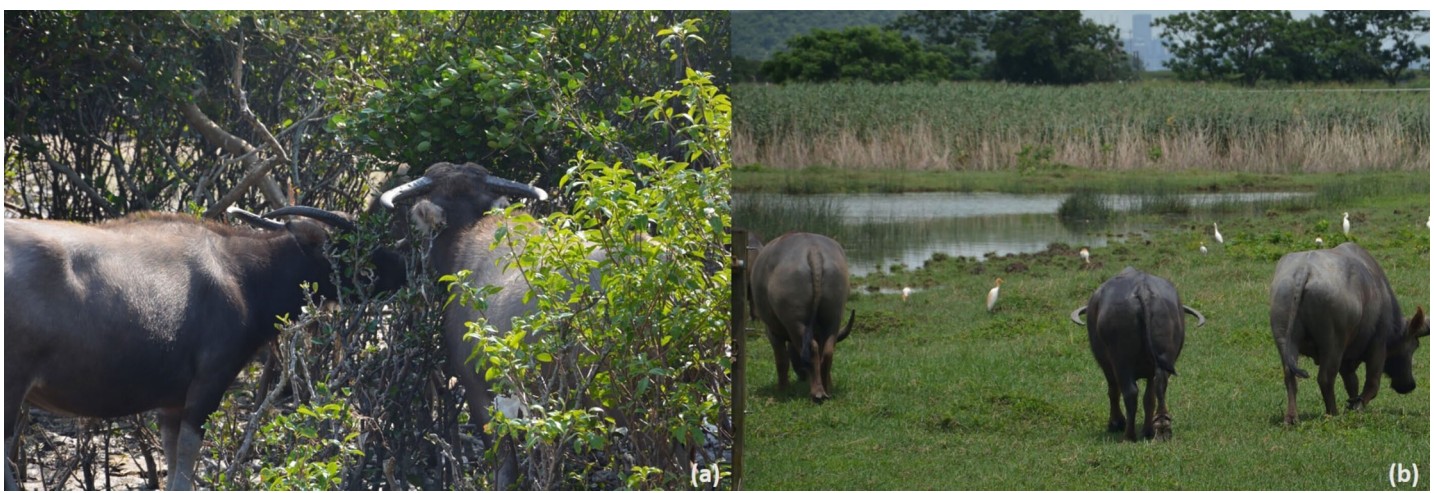

**Figure 4 (A) Feral buffaloes browsing and (B) walking in lowland marshlands in Hong Kong SAR, China.** Photo credit: George M. W. Hodgson.

absence of grazing, poor cleanliness *etc.* (*de Rosa et al., 2009b*). Buffaloes are often now kept in dairy farm sheds (5–10 m² per animal) with space availability in an adjacent outdoor farmyard (8–14 m² per animal), (*Napolitano et al., 2013*). In modern farming, confinement discourages natural behaviors such as wallowing (Fig. 3), and grazing (Fig. 5). Furthermore, space restrictions in more intensive systems results in the expression of undesirable behaviors such as teat suckling of other animals and excessive aggression, resulting in risk factors for injuries and poor welfare (*Napolitano et al., 2013*). Machine milking may involve appropriate equipment performance and calf detachment (removal of calf from mother), (*Napolitano et al., 2013*). The negative behavior of stockpersons, such as hitting, slapping, and yelling, can cause physical and emotional stress, leading to milk letdown problems, elevated heart rates, higher cortisol levels, and retained milk (*Rushen, De Passille & Munksgaard, 1999*; *Napolitano et al., 2013*). Other welfare issues associated with intensive buffalo farming include higher mortality levels, probably due to immune suppression caused by stress and poor housing conditions. Intensive farming also leads to the progression of abnormal behaviors, such as allosuckling and tongue rolling (*de Rosa et al., 2009b*).

Cattle were the first animals to be incorporated into intensive farming systems, with advancements like machine milking, and housing specifically developed for them (*de la Cruz-Cruz et al., 2019*). Currently, the same techniques are also being used for buffalo milk production. By gaining a deeper comprehension of buffalo behavior, physiology, and health, and making necessary adaptations (for example, housing, nutrition, and design of milking parlors) to their production systems, there is potential to enhance their welfare (*Deb et al., 2016*). It is important to recognize the significance of welfare throughout the entire production chain to enhance their welfare and production efficiency, as buffalo welfare extends beyond their ability to produce meat, milk and work as draught animals (*Mora-Medina et al., 2018*). Moreover, as buffaloes can adapt better to hotter than colder

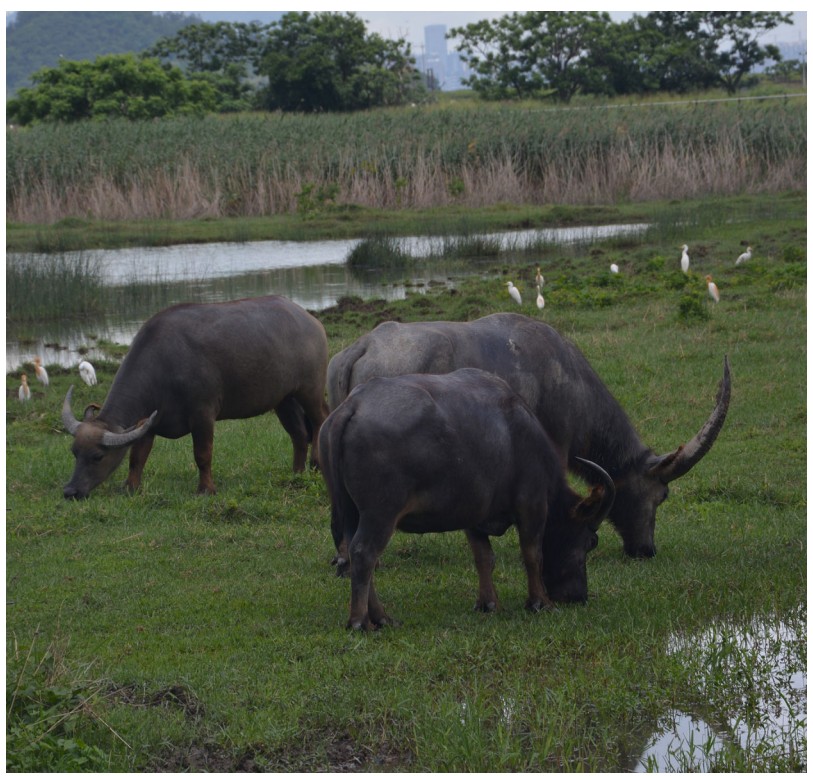

**Figure 5 Extensively grazing feral buffaloes in a lowland marshland in Hong Kong SAR, China.**
Photo credit: George M. W. Hodgson.   

environments (*de Rosa et al., 2009a*), this feature makes them an important future resource as global temperatures increase.

## Monitoring buffalo welfare

More attention needs to be focused on housing since it not only limits the natural buffalo behaviors, but it can also induce physiological changes related to stress due to limited space (*Mora-Medina et al., 2018*). Water pools in traditional farming systems have been phased out in favor of spray systems in order to minimize the time spent cleaning the udder, enhance milk hygiene, and decrease disease transmission during milking, such as mastitis (*Rosa et al., 2007*). However, buffaloes with access to a pool had a higher level of fertility and shorter calving intervals than those with shower stalls to help with body temperature regulation (*de Rosa et al., 2009b*; *Napolitano et al., 2013*). Providing buffaloes with approximately 36–40 m$^2$ space per animal along with access to vegetation and pools is important as those conditions allow them to express natural behaviors (for example, grazing and wallowing). Appropriate space per animal is also linked to increased milk production (*Rosa et al., 2007*). Therefore, to enhance buffalo welfare, housing should include water pools, adequate space, and access to natural grass (*de Rosa et al., 2009b*; *Napolitano et al., 2013*).

Animal cleanliness reveals information about their comfort, pleasure, and standard of care by stockpersons (*Winckler et al., 2003*; *Napolitano et al., 2013*). Dirty skin can

minimize antigerm and thermoregulatory features of the skin, as well as causing epidermal inflammatory disorders (*Winckler et al., 2003*; *Napolitano et al., 2013*). A buffalo cleanliness score has been developed but its repeatability was relatively low compared to cattle (*de Rosa et al., 2009b*). This could be due to their unique wallowing behavior, performed in order to get protection from biting insects and solar radiation. Thus, the appearance of mud on buffaloes can be potentially regarded as a positive trait, while a thick and compact layer of dung on their bodies may be considered a potential health risk (*Napolitano et al., 2005*, *2013*).

Buffalo body condition score (BCS) is used to estimate body energy stores by scoring the subcutaneous fat components of animals rather than live weight fluctuations (*de Rosa et al., 2009b*; *Napolitano et al., 2013*). This method entails assigning three distinct scores to each buffalo based on the categories of 'very lean,' 'regular,' and 'very fat.' BCS can be used to assess malnutrition and welfare, with an increased proportion of skinny animals associated with lower welfare (*de Rosa et al., 2009b*). Morphologically, buffaloes are more comparable to dual-purpose cattle (raised both for milk and meat purpose) than dairy cattle (*Napolitano et al., 2013*).

Damaged skin shows the effect of the surrounding environment on the body of an animal (*de Rosa et al., 2009b*; *Napolitano et al., 2013*). Contact with hard surfaces, drags against walls, feed shelves and cubicle dividers can cause skin damage (*Napolitano et al., 2013*). Thus, while assessing buffalo welfare, it is important to check for any swelling, callosity, and lesions on various body areas. These measurements reflect inappropriate management, feeding racks and housing design (*Napolitano et al., 2013*).

Higher buffalo mortality can result from poor management and bad welfare, leading to economic losses for farms (*Napolitano et al., 2013*; *Deb et al., 2016*). Important buffalo diseases include enteritis, respiratory issues, and mastitis, which lead to high mortality and culling. These are generally linked to nutritionally unbalanced feed fed to buffaloes (*de Rosa et al., 2009b*; *Napolitano et al., 2013*). Therefore, guidelines for assessing the buffalo welfare on farms must consider parameters that reveal their health and diseases. Animal health and disease information should be recorded on farms for animal welfare assessment, as mortality and culling due to illness and accidents are important indicators to quantify buffalo welfare (*de Rosa et al., 2009b*; *Napolitano et al., 2013*).

Buffalo calves that are kept in confined spaces (1.2 m$^2$ per animal), rest and lie for less time, and have more bent legs while lying as compared to those living in larger areas (2.1 m$^2$ per head) (*de Rosa et al., 2009b*). Animals also show decreased levels of lying idle, which is likely to represent a form of resting (*de Rosa et al., 2009b*). Cross-suckling occurs in both calves and adults, resulting in severe injuries to udders, prepuce and navel, as well as milk loss in dairy animals (*Napolitano et al., 2013*).

Amid decreased space allocation (up to 5–10 m$^2$ per animal) due to farm intensification, agonistic interactions are becoming an important problem for buffalo farming (*de Rosa et al., 2009b*; *Napolitano et al., 2013*). Buffaloes are also usually not dehorned and are free to engage in agonistic behaviors. However, social contacts are more likely inside a confined area, and flight possibilities of weak animals are greatly decreased (*de Rosa et al., 2009b*; *Napolitano et al., 2013*). Therefore, when the available space is too limited, the risk of

abnormal behavior, such as cross-suckling, greatly increases and can compromise welfare. (*de Rosa et al., 2009b*; *Napolitano et al., 2013*). Primiparous buffaloes are frequently mixed in with multiparous animals. Primiparous animals are usually have the lowest social status and therefore, they have a higher likelihood of udder and skin lesions due fighting for food, and water and space limitations (*de Rosa et al., 2009b*; *Napolitano et al., 2013*).

Overall, to establish a buffalo welfare monitoring system, it is essential to implement standardized methods for translating welfare related indicators into consumer-friendly information. Housing, body cleanliness, body condition score, lameness, normal social relationships, mortality rate, abnormal behaviors, and positive welfare indicators are essential measures to monitor buffalo welfare (*Hafez, Badreldin & Shafei, 1955*; *de Rosa et al., 2009b*). The most urgent research priorities for buffalo should be machine milking, housing, behavior, physiology, and standardization of farming and welfare monitoring systems (*de Rosa et al., 2009b*; *Napolitano et al., 2013*).

## DROMEDARY CAMEL WELFARE

There are approximately 38 million camels in the world, with 34 million of these being one-humped dromedary (*Oselu, Ebere & Arimi, 2022*). Camels are native to deserts and were first domesticated 4,500 years ago in the southern region of Arabia (*Oselu, Ebere & Arimi, 2022*). Dromedary camels are found in Australia, the Middle East, northern parts of Africa and Asia, and the Indian subcontinent (*Dioli, 2020*). Bactrian or two-humped camels (*Camelus bactrianus*) are found only in East, inner, and central Asia (*e.g.* Mongolia, China, Kazakhstan), northern areas of India, Pakistan, Iran, and eastern parts of Turkey (*Dioli, 2020*), however, they will not be the focus of this review section because only dromedary camel production has become intensive in recent years, with related welfare concerns.

Camels have advanced physiology that helps decrease water loss and allow them to withstand substantial dehydration (*Gebreyohanes & Assen, 2017*). They can go for several weeks without drinking water. Camel blood plays a crucial role in their adaptive response to the high environmental temperature as well as dehydration. Surprisingly, they can become dehydrated without affecting their blood viscosity. Even under extreme environmental conditions, their hemoglobin function remains unchanged (*Gebreyohanes & Assen, 2017*). The daily body temperature of a fully hydrated camel ranges from 36 °C to 38 °C. However, if dehydrated and subjected to excessive heat, body temperature can vary considerably by 6 °C to 7 °C, ranging from 34 °C to 41 °C (*Gaughan et al., 2010*; *Gebreyohanes & Assen, 2017*). Temperature variation is common in most animal species, but not to the same extent as camels. For example, when exposed to extremely hot climates, cattle would be able to cope with maximum 2°C to 4°C rises in body temperature (*Gaughan et al., 2010*; *Gebreyohanes & Assen, 2017*).

The thick coats of camels protect them from the extreme heat radiated by desert sand, and throughout the summer, the coat lightens in color, reflecting solar light and preventing sunburn (*Gebreyohanes & Assen, 2017*). Moreover, they have a thick tissue pad over the sternum known as the pedestal. When they lie down, the pedestal, along with other small padded contact points, raises the body from the hot ground surface and allows cooling air

to circulate beneath the body (*Gebreyohanes & Assen, 2017*). Furthermore, their hairy coat creates a sufficient buffer zone, and it separates the skin surface from surrounding climatic conditions. Coat thickness changes with the seasons to accommodate varying environmental conditions (*Gebreyohanes & Assen, 2017*). If possible, camels avoid lying in the sun; otherwise, they face the sun but try not to expose their entire body. They raise their sternum in the recumbent position to make a "plate-like" contour that enables air to circulate. Lying or standing, camels gradually shift their position throughout the day to align with the sun, minimizing the area exposed to direct solar radiation (*Eltahir et al., 2010*; *Gebreyohanes & Assen, 2017*; *Zappaterra et al., 2021*).

Camels are multipurpose livestock animals used for milk, hides, and meat and play an active role in cultural, recreational, agricultural, and sporting activities (*Ahmad et al., 2010*; *Padalino & Menchetti, 2021*). There is considerable interest in the nutritive value of camel milk and meat (*Ahmad et al., 2010*; *Padalino & Menchetti, 2021*). Camels are increasingly being farmed in intensive production systems with modern housing, feeding and milking methods (*Camelicious Farm, 2022*), in a similar manner to cattle. However, one of the key aspects is that we know almost nothing about camel welfare, with very little research. This may be because, until recently, farming was more extensive and less of concern as at least camels had some agency. They are also known as "tough" animals and able to live without food and water for long periods of time–maybe that leaves them more likely to be neglected. Presently, only three articles on camel welfare have been published, all focusing on semi-intensively managed camels with no mention of camels kept under extensive nomadic pastoralist operations (*Previti, Guercio & Passantino, 2016*; *Pastrana et al., 2020*; *Padalino & Menchetti, 2021*; *Dioli, 2022*). Welfare guidelines for intensively as well as semi-intensively farmed camels have been developed only recently. These do not correspond to the real-life of camels raised under nomadic circumstances (Fig. 6), and there are no specific guidelines for nomadic camel welfare (*Previti, Guercio & Passantino, 2016*; *Pastrana et al., 2020*; *Padalino & Menchetti, 2021*; *Dioli, 2022*).

## Camel production systems

There are three major camel production systems, *i.e.*, nomadic, sedentary, and modern farming (*Ahmad et al., 2010*; *Camelicious Farm, 2022*). These systems are primarily determined by number and purpose of animals, climatic conditions, the topography of the land, plant growth phenology, and water sources (*Ahmad et al., 2010*).

### Nomadic system

This system is found in several countries, including Pakistan (Fig. 6) and a typical nomadic herd constitutes about 24 camels (*Ahmad et al., 2010*). Lack of grazing and water shortages are the primary motivations for the camel herds to be moved from place to place (*Ahmad et al., 2010*). Camels are kept with other livestock (for example, sheep, goats and donkeys). Different livestock have different dietary preferences (*e.g.*, browsers and grazers). Thus, not only do they utilize a wider range of forage, but using diverse livestock also reduces the probability of total loss of animals for farmers, especially with challenging and variable climatic conditions. The movement of herds is a fundamental strategy for survival,

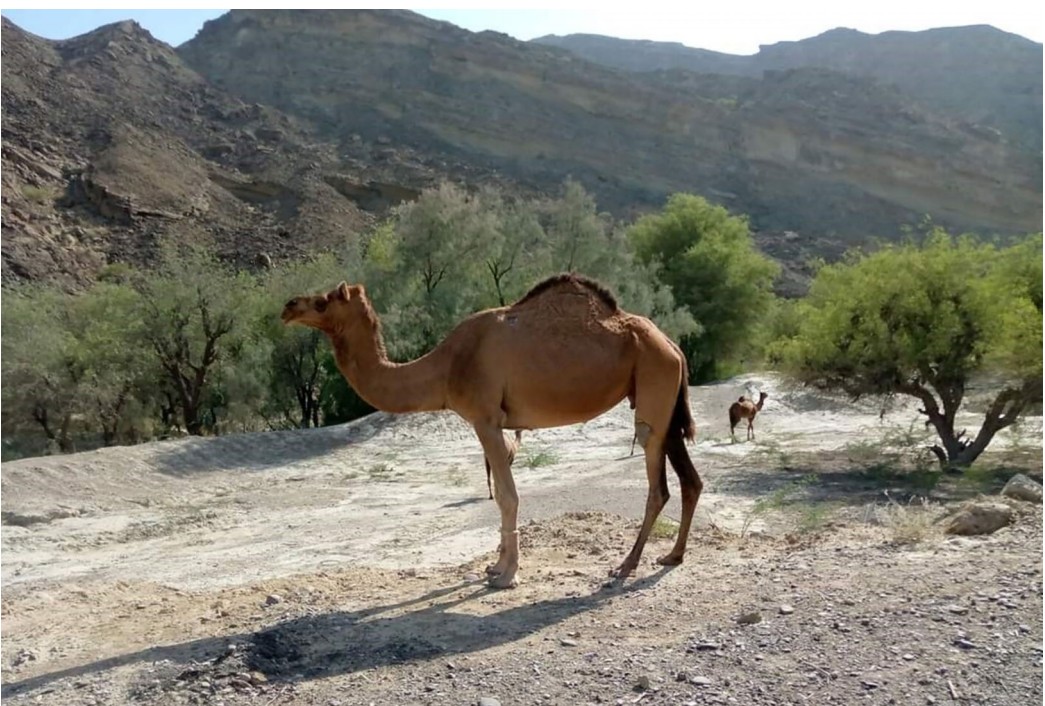

**Figure 6 Dromedary camels farmed using an extensive nomadic production system for milk and meat in Balochistan-Pakistan.** Photo credit: Muhammad Yaqoob.

including various types of migration (seasonal, short-distant, and long distant disaster migrations) (*Ahmad et al., 2010*).

### Sedentary system

In a sedentary system, camel herds settle in an area permanently, unlike in the nomadic system. A typical sedentary herd consists of 24 camels (72% female and 28% male) (*Ahmad et al., 2010*). Camel raising contributes one-third of household income and helps increase farm productivity in Pakistan, and there has been a steady decrease in the number of nomadic herds. Herds are gradually becoming sedentary by settling around permanent agricultural fields (*Ahmad et al., 2010*).

### Modern farming

Modern farming involves the intensive housing of hundreds of camels for various purposes such as milk and meat production. In the United Arab Emirates (UAE) alone, there are over 10,000 camels that are raised for milk. These camels are fed and milked using modern feeding and milking techniques (*Camelicious Farm, 2022*), similar to cattle. However, our understanding of camel welfare in modern farming is severely limited due to the scarcity of research conducted thus far (*Padalino et al., 2021*). This signifies a substantial gap in our existing knowledge that necessitates further investigation.

## Camel welfare issues and limitations for production

Nomadic camel welfare issues and limitations for production have been reported from various regions of Africa, the Middle East and South Asia (*Ahmad et al., 2010*; *Dioli, 2022*). Camels are often subjected to periods of hunger and starvation caused by erratic rainfall and decrease in forage availability. Moreover, drinking water points are usually several walking days away from the foraging areas. Camel night enclosures are often not appropriately built and routinely subjected to thermal as well as predator stresses. Furthermore, clinical veterinary services may be unavailable in nomadic pastoral areas. Diseases, parasites, and the pain and distress that can be associated with disease are thus a regular occurrence, adversely affecting camel health and welfare (*Dioli, 2022*).

Traditional husbandry methods (nomadic and sedentary systems), inadequate marketing of camel products, and calf mortality rates remain key barriers to overall camel production and welfare (*Faraz et al., 2021*). Moreover, inadequate marketing of camel products results in lesser income for camel owners, which forces them to compromise on their housing quality and other farm operations. This ultimately impacts the living standards and welfare of camels (*Faraz et al., 2021*). Ethno-veterinary medicines are commonly used as primary health care, influencing camels' overall health and production potential. It is crucial for concerned stakeholders and authorized institutions to evaluate the immediate needs, education, knowledge, and husbandry expertise of indigenous communities. This assessment will facilitate support for farmers and the enhancement of animal welfare in Low and Middle Income Countries (LMICs), (*Faraz et al., 2021*).

## Camel welfare assessment

Some camel welfare assessment guidelines have very recently been designed for intensive or farm-based animal production systems (*Kaurivi et al., 2020*; *Dioli, 2022*). However, they cannot be used directly to assess welfare issues for extensively managed or nomadic livestock (*Kaurivi et al., 2020*). Restraining methods, milking practices, weaning and fostering, calf acceptance, packing and riding practices, slaughtering methods and traditional ethno-veterinary treatments are some of the most common areas of camel production where their welfare may be compromised (*Dioli, 2022*).

Some of the most important factors to consider for herd level camel welfare assessment are proper feeding, housing, appropriate health and the ability to engage in normal behaviors (*Padalino & Menchetti, 2021*). There should be a proper feeding management framework for the herd for the provision and availability of nutritionally balanced feed. There must be a sufficient number of clean feeding points and feeding behavior of animals should be assessed (*Padalino & Menchetti, 2021*; *Dioli, 2022*). Comfortable housing should be provided for animals with appropriate bedding and space allowance. The behavior of animals needs to be considered while resting for welfare assessment (*Kaurivi et al., 2020*; *Padalino & Menchetti, 2021*). Animals must be free from injuries and diseases and there is a need to check for sick animals, type of injury and disease. Finally, there should be check regarding behavior of animal for welfare assessment, *e.g.*, social behavior, aggressive behavior, stereotypies *etc.* (*Padalino & Menchetti, 2021*).

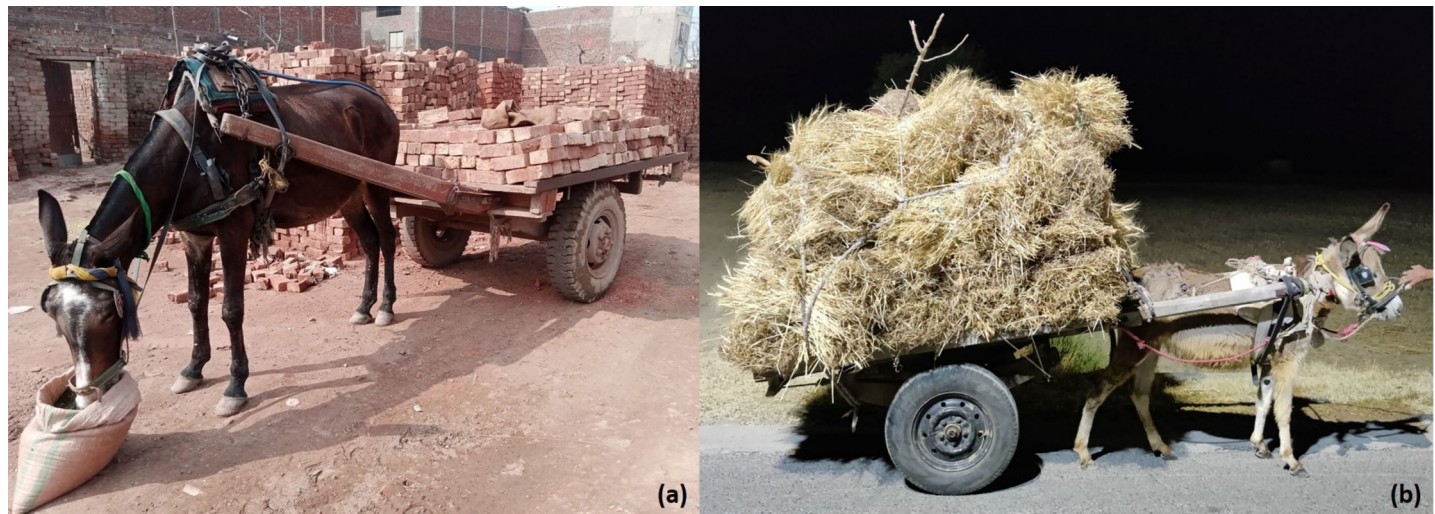

**Figure 7** (A) Load-pulling mule at an urban brick kiln production system in Pakistan. (B) Donkey pulling wheat crop in the rural agricultural production system in Pakistan. Photo credit: Syed S. U. H. Bukhari.

Individual animal level welfare assessment usually comprises a visual inspection to evaluate camel behavior and health status. Some of the most important points to check while assessing individual camel welfare are body condition score, presence of disease, resting behavior, and injuries due to tethering and hobbles (*Padalino & Menchetti, 2021*; *Zappaterra et al., 2021*). Moreover, camel feeding and drinking behavior should be examined. The good feeding and drinking principles are adequate nutrition and lack of prolonged thirst. Hunger and thirst can appear not only when feed and water are unavailable, but also when they are inaccessible or do not meet the behavioral and physiological needs of the animals (*Padalino & Menchetti, 2021*).

## DONKEY AND MULE WELFARE

Donkeys and mules have always had an important role in load-carrying, transport, draught, and agricultural production, especially in low- and middle-income countries (LMICs) (Figs. 7A, 7B), (*Bukhari, McElligott & Parkes, 2021*; *Bukhari & Parkes, 2023*). Donkeys are also of growing importance as a farmed species, and may also be kept as companion animals (*Davis, 2019*; *Bennett & Pfuderer, 2020*). The global donkey and mule population is approximately 50.5 and 8.5 million, respectively (*Norris et al., 2021*). They help sustain 600 million people worldwide, most of whom live in poor and marginalized communities (*Admassu & Shiferaw, 2011*; *Lanas, Luna & Tadich, 2018*). Working donkeys and mules are vital to people's economic and social wellbeing (*Valette, 2015*). Carts hauled by donkeys and mules are essential modes of transportation in most of these communities, and carting is a source of income for a large proportion of the population in developing nations (*Molla, Fentahun & Jemberu, 2021*). In LMICs, while motorized transportation has grown quickly during the last few decades, the use of donkeys and mules for local transportation of goods has remained unchanged (*Bukhari, McElligott & Parkes, 2021*; *Bukhari & Parkes, 2023*). These equids are crucial in the growth of agriculture and transportation as they provide power for ploughing and traction, playing an essential role

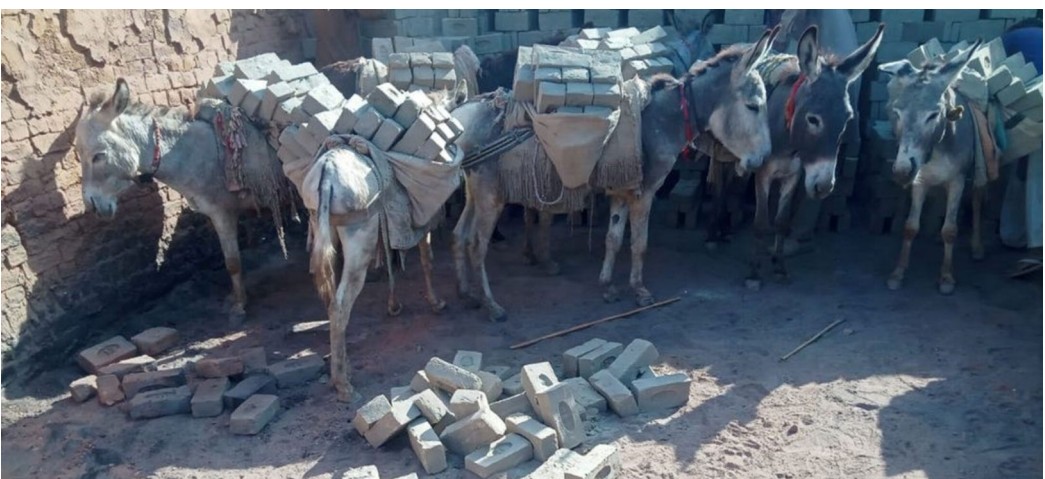

**Figure 8 Donkeys carrying bricks in Pakistan.** Photo credit: Syed S. U. H. Bukhari.

in the local economy. Donkey and mule social and economic contributions to rural income can be direct (providing transportation services) or indirect (*e.g.*, plowing the soil for crops) (*Valette, 2015*). Therefore, underestimating their contribution could have a negative impact on society. However, the economic role of working equids is often under-estimated or not assessed by policymakers (*Valette, 2015*).

## Welfare issues of donkeys and mules

The welfare standards of working donkeys and mules remain very poor (*Valette, 2015*; *Davis, 2019*). Some of the most common reasons for poor welfare are improper shelter, food, and water; high workload, poor handling (*e.g.*, driving and whipping) and harmful practices such as nostril slitting (*Davis, 2019*; *Bukhari et al., 2023*). In addition to this, a lack of supporting infrastructure (healthcare, good farriers, and saddlers), harsh environmental conditions, a lack of recognition in legal systems, marginalization, and absence of program enforcement also contribute to compromised working donkey and mule welfare (*Valette, 2015*; *Davis, 2019*).

Overloading of donkeys and mules is one of the many issues that may lead to reduced welfare, which is a global concern (*Bukhari et al., 2022*). Their most common welfare issues include overloading, poor physical condition, lameness, dehydration, heat stress, exhaustion, malnutrition, and other catastrophic injuries (*Valette, 2015*; *Bukhari, McElligott & Parkes, 2021*; *Khan et al., 2022*). The severity of welfare problems depends on the type of work the animals perform. For example, brick kiln work (Fig. 8) appears to be linked with increased welfare problems in working equids compared to other work areas, for example, agriculture (*Mitra & Valette, 2017*). The welfare of working equids can be improved through the collective actions of both equid-owning communities and supporting organizations (*Bukhari et al., 2023*).

Changes in consumer habits also have an impact on livestock welfare and behavior. Chinese medicine practitioners use gelatinous material extracted from the donkey hide as an elixir (*Bukhari et al., 2023*). Traditionally, this material, known as 'ejiao', is believed to

improve strength, treat anemia, and counter some psychological issues, including insomnia. Initially, the demand for donkey products was met by slaughtering large numbers of donkeys within China, resulting in a reduction in the Chinese donkey population. Increasing demand for donkeys has led to increased farming in China itself, as well as other countries such as Australia, Brazil, Kenya, Pakistan, and South Africa (*Davis, 2019*; *Tatemoto et al., 2021*). However, farming systems need to be improved to consider the welfare and behavioral needs of donkeys, as this mode of production is relatively new. Important behavioral needs include the freedom to graze and browse, as well as the opportunity to engage in normal behaviors such as forming social bonds with other animals (*Davis, 2019*; *Tatemoto et al., 2021*). Donkeys, being hindgut fermenters with a simple stomach, are adapted to "trickle feeding," which involves consuming small amounts continuously to meet their nutritional requirements. If donkeys are only fed twice a day (a common practice on intensive farms), their body condition and gastrointestinal health may suffer, and their nutritional needs may not be adequately met based on their stage of production (lactation, pregnancy, working, *etc.*), (*The Donkey Sanctuary, 2023*). Consequently, it remains uncertain whether intensive farming is suitable for this species, and evidence-based research is necessary to address these inquiries.

Every year, millions of equids are slaughtered (*Tatemoto et al., 2021*). Equine meat consumption is deeply rooted in the culture of certain countries, such as France and Japan, as well as in LMICs like Chile, where it is popular due to its affordability compared to beef (*Rubio Lozano et al., 2020*; *Fletcher et al., 2022*). However, there is a lack of consistency in equine slaughter guidelines worldwide, and published data on the topic are generally insufficient. This hampers their practical implementation and may compromise animal welfare, particularly in the moments leading up to slaughter (*Fletcher et al., 2022*). In LMICs, there is a scarcity of published studies that assess the suffering of animals during the slaughter process, especially for donkeys and mules (*Davis, 2019*; *Tatemoto et al., 2021*; *Fletcher et al., 2022*). This highlights the urgent need for research to establish an evidence base that can provide better guidance in this specific field. Furthermore, slaughterhouses should adapt their practices to accommodate the distinct needs of donkeys and mules, including access to food and water, efficient restraining, stunning, and slaughtering methods. However, there have been no studies conducted to examine the effects of these practices.

## Donkey and mule welfare assessment

Two specific welfare assessment tools exist for donkeys and mules. Firstly, the Equid Assessment, Research and Scoping (EARS) tool developed by the Donkey Sanctuary (*Raw et al., 2020*) and secondly, Standardized Equine Based Welfare Assessment Tool (SEBWAT) developed by the Brooke (*Browne, 2012*). The purpose of these tools is to provide an overview of the general welfare condition of animals, both individually and at a group level (*Browne, 2012*). The data generated by using these tools helps to determine the nature and prevalence of welfare issues occurring within an animal population. Secondly, they help to compare the welfare status between various locations and communities of

**Table 2 Some of the most important parameters to consider for welfare assessment of donkeys and mules (*Pritchard et al., 2005*; *Bukhari, McElligott & Parkes, 2021*; *de Santis et al., 2021*; *Bukhari & Parkes, 2023*).**

| Behavioral parameters | Health parameters |
|---|---|
| 1. General attitude of animal (alert, apathetic of depressed, dull facial expression, tail stillness, neck stiffness, head raised or hanging low and tense ears pointing backwards or to the side)<br>2. Response to observer approach (avoidance, aggression, or friendly approach) | 1. Body condition score<br>2. Skin lesions (wounds, scars, firing, branding *etc.*)<br>3. General health observations (diarrhea, ectoparasites *etc.*) |

interest. Third, they determine which groups of animals have the poorest welfare and the seasonal variation in welfare issues.

However, these protocols should be adapted to the context in which they are used since welfare issues differ in different parts of the world and even different parts of the same country (*de Santis et al., 2021*; *Bukhari et al., 2022*). For example, for working donkeys in LMICs resource-based protocols are not practical, while animal-based behavioral measures are particularly appropriate. Protocols should be easy to learn, low cost and rapid in their execution (*de Santis et al., 2021*). Behavioral measures for welfare assessment consist of the general attitude of donkeys and response to human approach (Table 2).

## LUMPFISH WELFARE

Atlantic lumpfish (*Cyclopterus lumpus*) are a teleost (bony) marine fish that are used as biological control agents in Atlantic salmon (*Salmo salar*) farming where their role is to act as "cleaner fish" to eat sea lice from the bodies of sea caged salmon (*Boissonnot et al., 2023*). Cleaner fish also include other species such as fishes from the Labridae family (ballan (*Labrus bergylta*), corkwing (*Symphodus melops*) and goldsinny wrasse (*Ctenolabrus rupestris*) (*Powell et al., 2018*; *Rey et al., 2021*), and these were first used in salmon aquaculture in the 1980s (*Bjordal, 1988*; *Deady, Varian & Fives, 1995*). However, currently only two species are farmed for this purpose, and these are ballan wrasse and lumpfish (*Brooker, Skern-Mauritzen & Bron, 2018*). In 2017 in Norway, approximately 29.7 million lumpfish (56%) and 1.0 million ballan wrasse of all stocked cleaner fish were hatchery-reared or farmed rather than being taken from the wild (*Fish Monitoring Centre, 2019*). Thus, the lumpfish is farmed to support the production of another farmed fish and their use signals a great loss of protein that could be used for human consumption. Given the much larger numbers of lumpfish (*C. lumpus*) produced, we focus on this species here. Their use poses unique ethical questions since lumpfish are employed as a biological tool to ensure the health and welfare of Atlantic salmon. For example, should their health and welfare be considered just as important as other production animals?

Lumpfish are found in the western and eastern Atlantic and juvenile lumpfish are often found in coastal areas clinging to rocks or kelp fronds using a suction disc on the ventral side of their body. After 1 year the adults move to the open ocean where they feed on plankton before returning 2–4 years later to coastal areas to spawn (*Imsland & Reynolds, 2022*). They are a non-shoaling species so do not form groups but there is evidence of

aggression and the formation of dominance hierarchies (*Rey et al., 2021*). Lumpfish feed on large planktonic organisms at the surface or mid-water and their diet includes zooplankton, fish eggs, and small crustaceans (*Davenport, 1985*; *Kennedy et al., 2016*). They are not considered strong swimmers and do require places to affix to using their suction disc to rest or avoid rough seas (*Geitung et al., 2020*).

## Lumpfish welfare issues

When used in Atlantic salmon production, lumpfish are placed into sea cages and are not only held with salmon but may also be held with wrasse also used as cleaner fish. Their diet in the sea cage consists of parasitic sea lice and they are confined to the shallow coastal conditions they are housed in. High mortalities have been reported for lumpfish and other cleaner fish species. One study documented that 27% of lumpfish died over a 12 week period conducted in Norway (*Geitung et al., 2020*). Cleaner fish mortalities in sea cages range from 18% to 48%, with some farms sustaining 100% mortality (or loss for unknown reasons) (*Nielsen et al., 2014*). Over 65% mortality of ~193,000 cleaner fish was recorded in 12 commercial salmon sea cages (*Bui et al., 2018*) and a recent industry report registered mortality at 42% (*Stien et al., 2020*). Given this high loss of life, it is crucial that we understand how to improve their health and welfare in these production systems.

A study exploring welfare, growth and survival of lumpfish demonstrated that these animals occupy shallower, cooler and less saline areas of sea cages (*Geitung et al., 2020*). One of the impacts of climate change is that oceanic temperatures are rising and thus lumpfish may be unable to seek cooler temperatures in the confines of a sea cage especially during heat waves. These animals are cold blooded and thus the environmental temperature has a substantial impact on their physiology. Heart rate measurements in response to increased temperature and exercise demonstrated that lumpfish have a relatively poor response with a low maximal heart rate and thus these fish are unlikely to perform well under conditions when temperatures rise (*Zrini, Sandrelli & Gamperl, 2021*). Preference studies show adult lumpfish prefer 6–7 °C but on land production of these animals is typically 10 °C (*Mortensen et al., 2020*). During a heatwave in Newfoundland, surface water temperatures rose from 10–12 °C to 18–19.5 °C. During this period the salmon within the cage reduced their depth to cooler areas (*Gamperl, Zrini & Sandrelli, 2021*). If lumpfish were present in this scenario, these temperatures would be much higher than their preferred range but they may also have to compete for space at cooler, lower depths with the relatively larger and more active salmon. Lumpfish used as cleaner fish are unable to choose their own habitat and may be confined in temperatures much higher than their temperature preference (*Hvas, Nilsen & Oppedal, 2018*; *Mortensen et al., 2020*). It is currently not known whether the thermal environment has an impact on the welfare of lumpfish in sea cages, but this is a crucial question that requires further study. Climate not only affects fish but can also lead to storms creating rough seas that might impair the welfare of lumpfish (*Zrini, Sandrelli & Gamperl, 2021*), especially if there is little for them to cling to within a sea cage. Other challenges in the sea cage environment linked to climate change concern interactions and dangers from other species such as the increasing prevalence of stinging jellyfish blooms. A recent report in the media stated that the

incidence of micro jellyfish blooms substantially impaired the welfare of Atlantic salmon within sea cages and could lead to high mortality (*The Herald, 2023*). However, it was not reported whether lumpfish were present or affected by these jellyfish bloom events; thus more information on how climate related changes in not only abiotic factors but also biotic or biological interactions with other species is needed.

Wild adult lumpfish are often caught during the spawning season to act as brood stock in captivity (*Powell et al., 2018*), where eggs are hatched by being held in incubators or hatching boxes. After hatching, lumpfish are typically fed commercial diets or live brine shrimp (*Artemia salina*) before being transferred to a growing farm, when they reach a size of 50mm (*Powell et al., 2018*). Lumpfish are typically reared on land in either tanks supplied by seawater from the marine environment in an open or flow through system or in a recirculating aquaculture system (RAS) (*Dahle et al., 2020*). The water used in RAS is either taken from the marine environment or is artificial seawater that is recirculated through a filtration system to keep water quality optimal. These are closed or semi-closed since to maintain salinity either new salt water or freshwater must be added and to maintain water quality some water has to be replenished at regular intervals to remove the accumulation of nitrate (end product of the denitrification of ammonia and nitrite). RAS is considered to be more environmentally friendly and sustainable, but it requires electricity or other power supplies to fuel electrical equipment *e.g.* water pumps and other filtration mechanisms. In 2018, 31 million lumpfish were farmed for use in sea cages in Norway. The number of cleaner fish farms in Norway mostly rear lumpfish and the numbers have increased from five to 31 from 2014 to 2019 (*Fish Monitoring Centre, 2019*). Few studies have explored the welfare of lumpfish from hatcheries to growing farms. One study reported housing conditions where the animals were transported from Iceland to Scotland and held in two flow-through larval rearing tanks. These tanks were 1.3 m$^3$ round PVC black tanks (150 cm diameter × 80 cm depth) at stocking densities of 10–15% (~12 kg/m$^3$). The temperature ranged from 8.2 °C (January) to 14.2 °C (October) suggesting a flow through system since temperature was not constant (*Rey et al., 2021*). No enrichment is described but it is possible these lumpfish could adhere to the bottom or sides of the tanks to meet their behavioral needs since wild individuals at this development stage are found adhering to rocks or algae. One study exploring the provision of shelters as enrichment within lumpfish tanks found that shelter provision reduced growth and body condition if food was delivered continuously rather than in pulses (*Johannesen, Joensen & Magnussen, 2018*). Reports have suggested that lumpfish prefer a black background rather than the typical blue or light grey aquaculture tank color (*Garcia de Leaniz et al., 2015*). There is still much to learn about the use of enrichment in lumpfish production.

Once lumpfish reach 20g they are deployed into Atlantic salmon sea cages (*Imsland et al., 2016*). They can be stocked at densities of 10–15% within the sea cages with studies reporting the use of artificial kelp (seaweed) to allow the lumpfish a surface to adhere to but also provide enrichment for the salmon within the cage (*Imsland et al., 2020*). Lumpfish spend a large proportion of time foraging during the day. They also have been observed affixed to floating seaweed or hovering under the seaweed (*Imsland et al., 2014a*). During the night, observations suggest they group together on smooth plastic or concrete

substrates (*Imsland et al., 2015a*). Thus, providing suitable resting substrates seems key in meeting their behavioral needs. Lumpfish are more active in sea cages with salmon present. This is not a result of aggressive interactions between the two species and so they appear to coexist without having negative impacts upon each other (*Imsland et al., 2014b*). Indeed, lumpfish do eat sea lice from the salmon body, and one could then infer a mutualistically beneficial relationship between the two species in this artificial context. However, it has been estimated that one third of lumpfish suffer mortality from starvation (*Powell et al., 2018*). The ingestion of sea lice does appear to increase over time since one study showed that the percentage of lumpfish feeding on sea-lice increased from ~15% at day 11 to ~35% at day 77 (*Imsland et al., 2015b*). Thus, identifying these parasites as food may be a learned behavior that could be "trained" to juvenile lumpfish before deployment into sea cages. Researchers found that younger lumpfish fed more readily on sea lice and showed a higher weight gain than larger size classes (*Imsland et al., 2016*). Indeed reports suggest that when lumpfish reach a size greater than 200 g their impact on sea lice is negligible (*Imsland & Reynolds, 2022*). Studies suggest variable success with use of lumpfish as a parasite control tool with 9% to 97% reduction of sea lice reported (*Imsland et al., 2018*; *Overton et al., 2020*). However, given the very high mortality rates of lumpfish one could pose the ethical question as to whether the use of one sentient being that has a ~33% of mortality in the first few weeks of being placed in a sea cage is worth it if the sea lice levels on another sentient being are not significantly reduced? To combat the loss of lumpfish through starvation supplementary feeding occurs *via* pelleted food for the lumpfish. Pelleted food is sporadically added to the cage and can be lost if not immediately consumed so one study explored the use of hanging food blocks where lumpfish has access to food over a longer time period (*Imsland et al., 2020*). These blocks did result in improved health and survival.

Another problem that arises within the sea cage that is of significant welfare concern is the lumpfish are held in close proximity with parasitized salmon. Sea lice do not just parasitize salmon but also attack any species of cleaner fish within the sea cage including lumpfish (*Garcia de Leaniz et al., 2022*). The welfare issues are three-fold: firstly, the damage that the sea lice do to the animal and secondly, the salmon are subjected to chemical and mechanical treatments used to remove sea lice that can lead to mortality in some cases thus there is a reduction in the lumpfish "diet" after treatments and thirdly; the lumpfish are subject to the treatments that affect their welfare. It has been reported that all animals in the sea cage undergo these treatments, and there is a notable rise in lumpfish mortality following mechanical delousing procedures (*Imsland et al., 2020*). The researchers attribute this increase to the mechanical delousing methods employed, suggesting that lumpfish should not be included in mechanical sea lice treatment.

## Lumpfish welfare assessment

Growth rates, body condition and survival (or mortality) rates have been used by several studies at different developmental stages of lumpfish including larval, on growing and adult lumpfish (*Powell et al., 2018*; *Dahle et al., 2020*; *Geitung et al., 2020*; *Rey et al., 2021*). Of course, the premise used here is that if the animals are growing, are in better condition

and are surviving between different experimental groups one can identify the optimal welfare conditions to house or farm lumpfish in. Whilst this does indeed have value, this does not address mental wellbeing. Researchers explored four methods of measuring body condition and found it did not relate to fin damage and suggested that in this case the tested methods were not useful to measure damage from aggressive interactions (*Rey et al., 2021*).

Wounds or sores on the lumpfish body can reflect general health, aggressive behavior and/or parasite or disease prevalence. Other measures can be used to assess health such as the incidence of cataracts, missing eyes or eye ulceration or swelling, spinal deformities or deformities to the suction disc (*Imsland et al., 2020*). Fin damage also indicates potential welfare issues and can be caused by poor water quality, injuries or aggression. By quantifying the extent of body and/or fin damage a health score can be attributed to lumpfish held in different contexts. Scientists derived a health score from the assessment of body wounds and fin damage which they propose can be used as a means of understanding the state of the animals as well as deciding when to euthanize (*Imsland et al., 2020*).

Lumpfish fin damage is seen in larvae and juveniles due to agonistic interactions in captive conditions or induced by stress (*Gutierrez Rabadan et al., 2021*). Fin damage prevalence appears to be reduced in the sea cage environment (*Brooker et al., 2018*; *Gutierrez Rabadan et al., 2021*). The main cause of fin damage in sea cages has been attributed to aggression but since space is greater and there is possibly less competition due to the abundance of sea lice, fin damage is seen at lower levels than in the hatchery or on growing farms (*Eliasen et al., 2020*; *Gutierrez Rabadan et al., 2021*). Fin damage scales can provide a means of quantitively assessing welfare. For example, scores can be made of the dorsal, tail (caudal) or anal fins. Recently researchers used a four point scale where a value of 0 = no damage, 1 = marginal biting or fin splitting, 2 = major distal fin ray loss, 3 = complete absence of fin (*Rey et al., 2021*). This study found that fin damage is most prevalent at young developmental stages due to high aggression. Fin damage poses a serious risk to welfare as it is likely to cause pain and can lead to compromised movement and infections in the damaged fin tissue (*Rey et al., 2021*). Therefore, it is imperative to explore strategies aimed at reducing aggression and minimizing fin damage.

Very little is known about the behavioral repertoire of lumpfish and so there is a dearth of knowledge on this subject (*Garcia de Leaniz et al., 2022*). First, we need to understand what is normal behavior before we can understand what is abnormal. Aggression is high in hatchery and on growing farms and this declines in the sea cage environment. From the limited study of lumpfish behavior in sea cages it appears as if most of the daylight period is spent foraging with short bouts of resting whereas lumpfish aggregate to rest on smooth surfaces during the night period (*Imsland et al., 2014a*). Five behavioral indicators have been proposed as useful for on farm assessment and they include loss of appetite, erratic swimming, lethargy, increased aggression and gasping (usually seen at the surface and may be an attempt to increase oxygen uptake) (*Gutierrez Rabadan et al., 2021*; *Garcia de Leaniz et al., 2022*).

## CONCLUSION

Animal welfare is a subject of important public debate, and it includes both emotional and physical wellbeing. In recent years, there has been a growing emphasis on promoting positive animal welfare states. However, research on animal welfare has been primarily focused on a limited number of livestock species. There are other important species to consider, such as buffaloes, camels, donkeys, mules, and lumpfish that deserve greater attention, especially considering the rising temperatures of our planet. Livestock such as buffalo, camels and donkeys have the ability to live and be used in farming in hotter climates and can be beneficial to humans. Therefore, we encourage researchers to look beyond the usual species when planning future projects and to pay attention to these neglected livestock species. While it is beyond the scope of this review to draw attention to all the neglected livestock species (*e.g.* ducks, and turkeys), they also deserve greater attention. We hope that this review will stimulate much-needed research on the needs of these and other neglected livestock species.

## ACKNOWLEDGEMENTS

Thanks to Claire Giraudet, George Hodgson, and Tania Perroux for their comments on an earlier version of this review and the editor and two anonymous reviewers for their useful comments.

### Funding

This project was funded by the City University of Hong Kong. The funders had no role in study design, data collection and analysis, decision to publish, or preparation of the manuscript.

### Grant Disclosures

The following grant information was disclosed by the authors:
City University of Hong Kong.

### Competing Interests

Alan G. McElligott and Rebecca S. V. Parkes are Academic Editors for PeerJ.
The remaining authors declare that they have no competing interests.

### Author Contributions

- Syed S. U. H. Bukhari conceived and designed the experiments, performed the experiments, analyzed the data, prepared figures and/or tables, authored or reviewed drafts of the article, and approved the final draft.
- Rebecca S. V. Parkes conceived and designed the experiments, performed the experiments, analyzed the data, authored or reviewed drafts of the article, and approved the final draft.

- Lynne U. Sneddon performed the experiments, analyzed the data, authored or reviewed drafts of the article, and approved the final draft.
- Alan G. McElligott conceived and designed the experiments, performed the experiments, analyzed the data, prepared figures and/or tables, authored or reviewed drafts of the article, and approved the final draft.

## Data Availability

This is a literature review.

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
