# Peer review of "The behavior and welfare of neglected species: some examples from fish and mammals"

_PeerJ, doi:10.7717/peerj.17149_

## Round 0.1 · original submission · Major Revisions

The two expert reviewers have provided very thoughtful and helpful suggestions for improving your manuscript so I will not reiterate their comments here. I will not do extensive line-by-line editing of my own because I agree with the reviewers that the manuscript requires extensive editing and reframing. Although one reviewer suggested “minor revision” in their recommendation, they too provided quite extensive suggestions for improvement. I think the paper needs a clearer focus and extensive reorganization before it can be considered for publication in PeerJ.

I do have several additional specific comments that I hope will be helpful as you revise your work.

In general, you need to develop your paragraphs more fully. The paper reads a bit like a list of facts and paraphrased quotations without providing a clear, cohesive framework and arguments with support from your own point of view.
Your section on Climate Change comes out of nowhere for me. I had assumed you were going to focus on direct effects of human interactions in farm settings. A paper on anticipated effects of climate change on animal welfare would be more unique and a welcome contribution to the literature. You could write a different paper framed specifically around this issue.

I don’t understand the third sentence of the abstract. Why “but” here? These clauses are not in contradiction. But, perhaps more importantly, what leads you to conclude that most decisions are motivated by self-interest? Although this is probably true, the statement negates all the efforts made by people to conserve and protect animals.

Your questions also seem too simplistic. Welfare is about more than what an animal “needs.” What about what it wants or what brings it pleasure? It probably needs veterinary care, but it would not enjoy most veterinary procedures.

Line 139, what do you mean by Global North bias?

I think it is likely that most of the publications on animal welfare originate from Western or Asian countries so a reflection of where the animals are studied and where the papers originate from may help place the disparities in context.

I think you should provide a stronger rationale for pairing species in your over-under studied pairings.

Reviewer 1 ·

Basic reporting

This manuscript provides an overview of some welfare issues associated with several livestock species that are not traditionally assessed in animal welfare research. It provides some information gleaned from both scientific papers and websites. As such, a range of different papers are cited in the work. It should be noted, however, that often citations are missing, including where key biological facts or population information is provided. It is essential that the sources are credited wherevers these facts are provided.
The raw data are currently not provided. The data (e.g. papers sampled) would improve repeatability and therefore scientific validity of the study.
The review covers a range of species. However, the selection criteria is somewhat ambiguous (see study design for more information). While there is no recent review of welfare for this selection of species, the justification of the work needs to be made clearer.

Experimental design

The article topic fits within the wider remit of PeerJ and could provide useful information with broad application to the farming industry globally. However, at current the study design limits the scientific quality and application of the work.
Focusing on the review methodology, the selection of species for focus appears unscientific. For example, species appear to have been selected because the authors believed they were understudied. This is a circular argument which also means that many species (which may have an even more pressing need for study) could have been ignored. A more appropriate, scientific selection criteria would involve identifying all farmed species and their population numbers, and then identifying the research output per species.
The methods for identifying species also need to be clearer. For example, exclusion criteria have not been provided and the inclusion criteria are very broad (encompassing all papers identified using the search engine). I ran a couple of searches using the criteria stated in the methods and found many more papers - so exclusion criteria need to be made far clearer. I would also strongly recommend stating what information was searched for in the sampled papers, or how papers were used.
The paper is structured on a species basis, but the subheadings for each species are not consistent. The manuscript would be improved with a more logical structure.

Validity of the findings

Currently, the manuscript does not provide novel information, but rather collates information from existing sources. Often, sweeping statements are made regarding housing and husbandry and welfare, and it is not clear there is sufficient evidence to support the conclusions drawn. It is not clear that the key welfare concerns have been identified, or that they are supported by the literature. As such there needs to be a deeper and more critical evaluation of the literature.
The authors have made a valid point that more research is required in order to improve welfare for the stated, farmed species. However, for this review to stimulate further research, the future avenues in need of research should be more clearly signposted.

Additional comments

If the work is to be reconsidered, the following areas should be addressed in full:
1. Provide an objective, repeatable method of selecting species for study.
2. Provide a clear and repeatable method of identifying papers
3. Explain what information is being extracted from each paper, and ideally provide the raw data
4. Provide a critical evaluation of current welfare and future directions for improving welfare research

Annotated reviews are not available for download in order to protect the identity of reviewers who chose to remain anonymous.

Reviewer 2 ·

Basic reporting

The manuscript 89181v1, titled “The welfare of the neglected species: some examples from fish and mammals” considers the welfare of four under studied species (buffaloes, camels, donkeys and mules, and lumpfish) using available literature. This is an interesting and worthwhile undertaking, and in particular, the lumpfish section was very well considered. The results/discussion presented in the manuscript are of interest and within the scope of the journal, however, some consideration of the comments below may improve the publication – please see comments below for consideration.

• the manuscript uses clear, professional English.
• sufficient context is provided with the exception of the introduction – see comment below.
• the overall structure, tables, figures are appropriate – see comments regarding subheadings and summary table of literature considered.
• the review is of interest and within the scope of the journal, and the topic has not been recently reviewed.

Introduction: The introduction feels somewhat superficial and would benefit from some context around why the key topics (L45 - 92) are considered and how they relate to the main body of your review. For example, are these the frameworks and/or key issues you will utilise when reviewing the literature concerning your 'neglected' livestock species? Also, for consistency should the subheadings be numbered 1.1, 1.2, etc.
• L61: What is the Five Freedoms (not just what are each of the freedoms) and how has it been used? E.g., the first widely accepted evidence-based framework to capture the key aspects of animal welfare (required to minimise welfare risk) in one model. In the past, it was widely used as a framework for consideration of animal welfare, particularly in terms of farm animals.
• L76 – 77: “The model provides a more comprehensive method for assessing welfare”. Many would argue that the Five Domains model is a framework to consider animal welfare rather than an assessment tool. It enables systematic consideration of all sources of possible
• L110: a subheading here may be beneficial to indicate a change in topic (e.g., aims/objectives).
• L110: it may be useful to briefly outline what you mean by ‘neglected’, e.g., subjected to limited scientific investigation, limited available literature, etc.
• L116 – 117: “have enormous potential to assist humanity in dealing with the challenges of climate change”, how? Please briefly expand.

Experimental design

• the manuscript is within the Aims and Scope of the journal.
• the methods are described with sufficient detail and information to replicate, although could be improved by the use of a table or flowchart – please see comment below.
• for the most part, sources are adequately cited – please see individual line comments below.
• the review is organised logically into coherent paragraphs and subsections, however may be improved by the use of common subheadings within each species section – see comment below.

Introduction
• L85 – 86: please add supporting references.
• L86 – 87: “However, promotion of the positive animal welfare states has become increasingly
• important in animal welfare over the past 20 years (Briefer, Tettamanti & McElligott, 2015)”, please add further supporting literature – if you are suggesting that positive animal welfare is becoming increasingly more important over the last 20 years you need to provide further evidence, including more recent literature.
• L103 – 107: please add supporting references.

Methods
• L124 – 125: “then paired them with species that that have been the subject of many studies over the years”, does this comparison simply relate to number of publications or more detailed comparison.
• Although this is not a systematic review, some form of table or flow chart which outlines the methodology (including search engine, key terms, selection criteria, papers identified (rejected and dismissed), etc. may improve clarity of the methods section.
• A summary table, briefly outlining the papers identified and considered for each species may add value.


Results/Discussion (species sections)
Clarity and linkage between species sections may be improved by using some consistent common subheadings within each species section. For example, production/housing systems, key welfare issues, welfare assessment, could be utilised within each species section. This may also help reduce repetition/overlap, for example, in the buffalo section. It may also be beneficial to conclude each section with a ‘priorities’ subheading (or something to that affect) that covers the main points you wish to raise regarding each species and link back to aims/objectives.
• Buffalo: as you have done for subsequent species, it would be beneficial to briefly outline the main reasons why buffalo are farmed, i.e., for meat, milk, use as working animals, etc.
• L174: need to provide a reference rather than [10].
• L178: Why are ‘natural’ behaviours important for welfare. Perhaps this is one element that needs to be considered further in the introduction.
• L185 – 186: please provide supporting references.
• L192: I presume you are referring to ‘negative’ stockperson behaviour? Perhaps provide some relevant examples for buffalo.
• L207 – 211: There is literature on the development of welfare assessment parameters and tools for buffalo (e.g., Winckler et al., 2003; De Rosa et al., 2005; Napolitano et al., 2005; De Rosa et al., 2015; Alam et al., 2020, etc.), it would be beneficial to briefly consider how past research can be applied and what is still needed.
• L226 – 227: please add supporting references.
• L235 – 236: please add supporting references.
• L241: please add supporting references.
• L247 – 248: please add supporting references.
• L252 – 253: “Information on animal health and diseases should be recorded on other farms for animal welfare assessment”, please provide some clarity as I don’t understand what point you are trying to make here.
• L262 – 264: is there any supporting literature?
• L275 – 279: text is in italics.
• L325: are their two or three main production systems (nomadic, sedentary and modern)?
• L369 – 370: please add supporting references.
• L375 – 382: any supporting literature?
• L413 – 415: please add supporting references.
• L436 – 437: What potential adaptions may be needed for donkeys?
• L438 - 439: What are these different needs and what practices may need to change?
• L546: remove full stop after 'to'.
• L567: please add supporting references.
• L576 – 579: a very good question.

Validity of the findings

The conclusion reads like a summary and could be more impactful if a few key outcomes of the review were reinforced and linked back to the manuscripts aims/objectives. This may also be helped by the addition of concluding sections for each species.

Additional comments

No additional comments.

---

## Round 0.2 · Minor Revisions

Thank you for your extensive revision in response to the reviewers’ comments. I appreciate the time and effort and think that the paper now flows better and the rationale for your focus is clearer. The additional references have helped to strengthen your arguments. However, the reviewers have some remaining suggestions that I hope you will consider before I can render a final decision. In addition:

Delete the extra . on line 123 (PDF version)
Shouldn’t the paragraph beginning on line 253 be moved to under Section 3.2?
The lines from 285-287 are redundant. Cross-suckling (on line 316, it is cross-sucking – please check spelling) is also mentioned multiple times. Please check throughout to eliminate redundancy. Similarly, you mention twice that lumpfish are not strong swimmers (lines 558 and 591). Line 626 – this was also stated earlier.
Lines 305 and 658, 664 change “which” to “that.”
Lines 597, 663 a ; is needed before ‘thus.’
Line 603, transferred to an ? (missing word?)
Do not use “since” in place of ‘because.”
Line 616, delete “on.”
Line 670, delete “a” before “several.”

Reviewer 1 ·

Basic reporting

The authors have provided an updated version of their manuscript and have showed considerable evidence of revisions in the work. There is good evidence of a wider range of papers in their reference list and these are well used in the text. The work is now better organised, with a clearer structure to the species-specific sections. The work covers several species that have featured less commonly in the wider literature. Given that these species do not feature as often, it is useful for a review to cover these taxa. The introduction initially introduces welfare but the definition still requires some clearer evaluation on what it incorporates. A rationale has been provided, but a little more clarity is needed on taxonomic selection.

Experimental design

The topics covered in this manuscript fit well within the wider remit of PeerJ. However, the methodology is still somewhat confused. The selection of species, for example, still remains unclear. Why were these specific taxa selected when other taxa (e.g. quail, ducks) which are held in great numbers, were not? Was the selection based on numbers of animals in captivity? Or number of locations holding the animals? Without this justification the selection seems arbitrary.
The arguments for comparing the species selected also seem somewhat arbitrary. While the phylogenetic argument is appropriate for buffalo-cattle and donkey-horse, it is tenuous for sheep-dromedary, and inappropriate for salmon-lumpfish (which have only a taxonomic class in common). There needs to be clearer justification for the comparisons.
The methods is something of a concern as it appears that the authors have just searched for ‘buffalo welfare’ and then reported the number of papers that this has pulled up. The assumption is that all of these papers actually discuss buffalo welfare: the reality is that many of these papers may be discussing other topics. This section needs to be either made clearer (i.e. how many of the papers identified are actually about buffalo welfare) or this information should be removed to avoid reporting spurious results.

Validity of the findings

Some key welfare assessments have been provided in the work. These have been developed since the initial review of the work. There are some interesting points drawn in the work. However, the gaps in the literature and the selection of species has resulted in some very specific findings. Some of the conclusions drawn are a little sweeping – these should be adjusted so that they apply to the species actually researched in the work.

Additional comments

While the manuscript is improved since the initial submission, revisions are still required. There needs to be a much clearer explanation of species selection, and the methods need to be either strengthened or the corresponding results removed. The work would also benefit from a full proof read to remove spelling and punctuation errors (e.g. including scientific name use).

Annotated reviews are not available for download in order to protect the identity of reviewers who chose to remain anonymous.

Reviewer 2 ·

Basic reporting

I acknowledge and appreciate the authors’ responses to my previous comments, and believe the changes made to address comments by the editor, reviewer 1 and myself have improved the manuscript. As previously noted, this review is an interesting and worthwhile undertaking, that highlights a number of species where further animal welfare research is needed. The results/discussion presented in the manuscript are of interest and within the scope of the journal, however, some consideration of the comments below may improve the publication – please see comments below for consideration.

• the manuscript uses clear, professional English.
• sufficient context is provided with the exception of a clear aim/objective of the review – at present it is a vague aim to highlight ‘neglected’ species. What does this achieve? By highlighting neglected species, would it not also be worth identifying specific research priorities that should be addressed within each species? As noted by reviewer 1, for this review to stimulate further research, the future avenues in need of research should be more clearly outlined.
• the overall structure, tables, figures are appropriate – the addition of figure 1 and consistent subheadings within each section improves the manuscript.
• the review is of interest and within the scope of the journal, and the topic has not been recently reviewed.

Experimental design

• the manuscript is within the Aims and Scope of the journal.
• the methods are described with sufficient detail and information to replicate – this has been improved in the revised manuscript with additional text and the addition of Figure 1. However, I think it is also important to provide some rational for the timeframe searched. Why is your search restricted to literature published post January 2000?
• for the most part, sources are adequately cited – please see individual line comments below.
• the review is organised logically into coherent paragraphs and subsections – the addition of consistent subheading within each section has improved organisation and readability of the manuscript.

*Please note line numbers refer to PDF version of the manuscript rather than the word version with track changes.

Introduction
• L123 – 124: Clearly outline aim/objective of the review.

Methods
• L136 – 138: Justification for timeframe search? Why is your search restricted to literature published post January 2000?
• L154 – 156: ‘Here, we use the phrase "Global North bias" to refer to the tendency for animal welfare studies to primarily focus on animals of interest in the Global North’. Evidence? Has this been previously quantified?
• L165: lowercase ‘s’ for Sheep.
• L178 - 180: Please cite reference(s).


Results/Discussion (species sections)
As noted in the previous review of the manuscript. It would also be beneficial to conclude each section with a ‘research priorities’ subheading (or something to that affect) that covers the main points you wish to raise regarding welfare research priorities for each species and link back to aims/objectives.
• L235 – 236: please cite reference(s).
• L246 – 248: What types of adaptions to production may be made?
• L248 – 250: Rather than broadly suggesting a greater understanding is needed, it would be more useful to identify more specific issues/factors that need to be investigated.
• L321: So as per earlier comment, where should research priorities be focused for buffalo?
• L357 – 358: please cite reference(s).
• L412: What are the traditional husbandry procedures and inadequate marketing (is this a spelling mistake?)? And how do they impact camel welfare?
• L415 – 416: How can this be done? Again, these broad statements are not likely to facilitate/encourage research – need to identify key aspects to be investigated.
• L482: ‘collective actions’ – what are these actions?
• L484: please cite reference(s).
• L512: ‘adapt their practices’ – what adaptions could/should be made? Do the effects of these adaptions on donkey welfare need to be examined?
• L698 – 701: please cite reference(s).

Conclusions
• L720: This is where some key research priorities could be given for each of the species you have reviewed. This provides clear direction for future research rather than a general statement that research is needed.

Validity of the findings

• The review would benefit from the authors identifying specific research priorities that should be addressed within each species, rather than more general and broad statements that research is needed on behaviour and welfare. As noted by reviewer 1, for this review to stimulate further research, the future avenues in need of research should be more clearly outlined – this remains true for the resubmission.
• The conclusion revision is useful, however if you identify specific research priorities for each species, these should be added to the conclusion.

Additional comments

No comment.

---

## Round 0.3 · accepted · Accept

I just have one small comment. I think you misunderstood my previous remark about the term "cross-sucking" on line 308. Did you also mean "cross-suckling" here? If so, please correct during proofing.

There are a few other inconsequential grammatical issues that you could fix during proofing:

Line 539, "differs" should be "differ."
Line 559, add a , after "produced." Please check throughout and add commas after clauses and words like "thus."
Line 601 "which" should be "that."